# Antimicrobial Susceptibility of Commensal *E. coli* Isolated from Wild Birds in Umbria (Central Italy)

**DOI:** 10.3390/ani13111776

**Published:** 2023-05-26

**Authors:** Laura Musa, Valentina Stefanetti, Patrizia Casagrande Proietti, Guido Grilli, Marco Gobbi, Valeria Toppi, Leonardo Brustenga, Chiara Francesca Magistrali, Maria Pia Franciosini

**Affiliations:** 1Department of Veterinary Medicine, University of Perugia, 06126 Perugia, Italy; laura.musa@studenti.unipg.it (L.M.); patrizia.casagrandeproietti@unipg.it (P.C.P.); valeria.toppi@studenti.unipg.it (V.T.); leonardo.brustenga@studenti.unipg.it (L.B.); mariapia.franciosini@unipg.it (M.P.F.); 2Department of Veterinary Medicine and Animal Sciences, University of Milan, 26900 Lodi, Italy; guido.grilli@unimi.it; 3Istituto Zooprofilattico Sperimentale dell’Umbria e delle Marche “Togo Rosati” (IZSUM), Via G. Salvemini 1, 06126 Perugia, Italy; m.gobbi@izsum.it (M.G.); c.magistrali@izsum.it (C.F.M.)

**Keywords:** wild birds, wildlife rescue centre, Central Italy, *Escherichia coli*, antimicrobial resistance, ESBL, beta-lactams

## Abstract

**Simple Summary:**

The scientific community has recently turned its interest to wildlife, including birds, as a potential marker of environmental antimicrobial resistance. The aim of this work was to investigate the antimicrobial susceptibility of 100 commensal *Escherichia coli* strains isolated from wild birds admitted to the Veterinary Teaching Hospital of Perugia (Central Italy) and the possible presence of extended-spectrum beta-lactamase (ESBL)-producing *E. coli* and *Salmonella* spp. Antimicrobials have been selected on the basis of their relevance for public health. The majority of the birds investigated were nocturnal and diurnal raptors and came from “WildUmbria”, a wildlife rescue centre in Central Italy. The initial clinical assessment revealed injuries mainly due to traumatic events. The *E. coli* isolates displayed significant resistance (*p* < 0.001) to ampicillin (85%) and amoxicillin/clavulanic acid (47%), which are widely used in veterinary and human medicine. Resistance to ciprofloxacin, cefotaxime, and ceftazidime showed values of 18%, 17% and 15%, respectively. Eight out of the hundred *E. coli* isolates (8%) were ESBL and seven displayed a multidrug resistance profile. *Salmonella* spp. was not isolated. Resistance to beta-lactams in all multidrug-resistant *E. coli*, including the presence of third-generation cephalosporins, highlights the need to increase wildlife monitoring studies to assess the potential risk to public health.

**Abstract:**

The role of wildlife, including birds, in antimicrobial resistance is nowadays a speculative topic for the scientific community as they could be spreaders/sources of antimicrobial resistance genes. In this respect, we aimed to investigate the antimicrobial susceptibility of 100 commensal *Escherichia coli* strains, isolated from wild birds from an Umbrian rescue centre and admitted to the Veterinary Teaching Hospital of Perugia (Central Italy) mainly for traumatic injuries. The possible presence of *Salmonella* spp. and ESBL-producing *E. coli* was also estimated. The highest prevalence of resistance was observed for ampicillin (85%) and amoxicillin/clavulanic acid (47%), probably due to their extensive use in human and veterinary medicine. Seventeen out of the one hundred *E. coli* isolates (17%) displayed a multidrug-resistance profile, including the beta-lactam category, with the most common resistance patterns to three or four classes of antibiotics. Resistance to ciprofloxacin, cefotaxime and ceftazidime exhibited values of 18%, 17% and 15%, respectively. Eight out of the hundred *E. coli* isolates (8%) were ESBL and seven showed multidrug resistance profiles. *Salmonella* spp. was not isolated. Resistance to third-generation cephalosporins, also detected in long-distance migratory birds, suggests the need for monitoring studies to define the role of wild birds in antimicrobial resistance circuits.

## 1. Introduction

Nowadays, antimicrobial resistance (AR) is considered a global threat that requires a One Health approach, involving animals, humans and the environment. The scientific community has recently turned its interest to wildlife, including birds, as potential markers of the level of environmental contamination by antimicrobial resistance genes (ARGs) [1]. Numerous reports have documented AR in wild birds, such as cormorants, birds of prey, gulls, doves and passerine birds [2,3,4].

Wild animals are not usually treated with antimicrobials, although they can acquire resistant microorganisms from different ecological niches. Soil represents a biodiverse habitat for several bacteria hosting ARGs, so it becomes an important link between humans and animals [5]. Manure and sewage, widely used as fertilizers in agriculture, can be affected by the use of antibiotics in farms and can, in turn, be a source of resistance for other animals, including wildlife, which feed on them [6]. Nelson et al. [7] demonstrated that gulls have been found to spread the same strains of *Escherichia coli* isolated from rubbish and waste-water treatment plants. Moreover, the sea can also be a source of antimicrobial resistance because of the high antimicrobial use in fish and seawater farming, which is often poorly regulated in some countries [8]. Furthermore, several species of fish, contaminated by AR bacteria/genes from humans, agricultural sewage and the aquaculture industry, can travel long distances and enter the human food chain [8,9].

The geographic location also plays an important role in influencing the prevalence of AR in wildlife, depending on the livestock and human populations and the degree of isolation of the area [10].

Other ARGs are thought to be ancient elements that evolved before the use of antibiotics [11], and intrinsic ARGs are present in most organisms [12,13,14]. Laborda et al. [15] also suggested that some mobile ARGs of clinical relevance could be present in wildlife as a result of selective pressure prior to the antibiotic era. In this scenario, migratory birds can spread resistant organisms across different geographical areas, given their ability to cover long distances in a short time along their travel routes [5]. The presence of AR plasmids has been reported in birds living in the Arctic region, which is considered an environment with low or absent antibiotic contamination [16] as a result of human pollution, although the possible spread by bird migration cannot be excluded [16,17]. In this scenario, *E. coli* and *Salmonella* spp. are considered to be microorganisms in which the selection of resistance genes has occurred rapidly over the years, due to the widespread use of antimicrobials in farming [18]. Thus, they may be able to play a consistent role in ARG co-circulation between the environment, animals and humans. Moreover, the progressive appearance of extended-spectrum beta-lactamase (ESBL)-producing *E. coli* and *Salmonella* spp. poses a threat to public health with regard to their therapeutic treatment possibilities in humans [19]. Food-producing animals, especially poultry, are considered the main source of ESBL-producing bacteria for humans via direct contact and/or consumption of contaminated meat products [19]. ESBL-producing *E. coli* was first isolated in wild birds in 2006 in Europe [20]. Since then, there have been several reports on their presence in numerous species of wild birds in European countries [21,22,23]. The main goal of this work was to investigate the susceptibility of commensal *E. coli* isolates to the antimicrobials of clinical relevance for Public Health in wild birds admitted to the Veterinary Teaching Hospital of Perugia (Central Italy). The possible presence of ESBL-producing *E. coli* and the *Salmonella* spp. was also evaluated.

## 2. Materials and Methods

### 2.1. Sampling

A total of 100 cloacal swabs were collected from different wild bird species on arrival at the Veterinary Teaching Hospital (Department of Veterinary Medicine, Perugia, Central Italy) in order to take samples before any possible antimicrobial therapies. Cloacal swabs were collected, preserved in a transport medium, a buffer solution with carbohydrates and peptones (Microbiotech, Maglie, Lecce), and then sent to the laboratory at 4 °C for bacteriological examination. The majority of the investigated birds were raptors (Table 1) and came from “WildUmbria”, a wild animal rescue centre in Central Italy. The initial clinical assessment revealed injuries mainly due to traumatic events (hunting wounds, road traffic, predation by dogs and cats or by other animals, etc.).

### 2.2. Microbiological Analysis

#### 2.2.1. Isolation and Identification of *E. coli*

In order to isolate *E. coli*, samples were placed in buffered peptone water (BPW) (Thermo Fisher Scientific, Rodano, Milan, Italy), which is a pre-enrichment medium at a ratio of 1:10, and then incubated for 18–24 h at 37 °C under aerobic conditions. A 0.1 mL of solution was taken from each sample diluted in this way, seeded on MacConkey agar (Thermo Fisher Scientific, Rodano, Milan, Italy), and incubated at 37 °C for 18–24 h under aerobic conditions. All the colonies with typical *E. coli* morphology were selected and identified using a MALDI-TOF MS instrument (Microflex LT Smart Biotyper with FlexControl Biotyper 3.4 software, Bruker Daltonics, Bremen, Germany). The isolates were identified as belonging to the *E. coli* species when they were lactose-fermenting and the score value for identification after MALDI-TOF analysis was above 2.

#### 2.2.2. Isolation and Identification of *Salmonella* spp.

One ml of the sample from the transport medium was inoculated in 9 mL of BPW and incubated at 37 °C for 18–24 h under aerobic conditions. Then, 0.1 mL of the pre-enriched inoculum was transferred to 10 mL of Rappaport-Vassiliadis Broth (Thermo Fisher Scientific, Rodano, Milan, Italy) for selective enrichment and incubated at 42 °C for 24 h under aerobic conditions. Subsequently, a loopful (10 µL) of inoculum was streaked on chromogen plates for *Salmonella* spp. (Liofilchem s.r.l, Roseto degli Abruzzi, Teramo, Italy) and incubated at 37 °C for 18–24 h under aerobic conditions (ISO-6579) [24].

### 2.3. Antibiotic Susceptibility Testing and ESBL-Producing E. coli Detection

To assess antimicrobial susceptibility, all the *E. coli* isolates were analysed using Kirby-Bauer disk diffusion method performed according to the CLSI document VET01 [25] for the following antibiotics: imipenem (IMP) (10 μg), cefoxitin (FOX) (30 μg), cefepime (FEP) (30 μg), cefotaxime (CTX) (30 μg), ceftazidime (CAZ) (30 μg), ampicillin (AMP) (10 μg), amoxicillin/clavulanic acid (AMC) (30 μg), chloramphenicol (CHL) (30 μg), azithromycin (AZM) (15 μg), nalidixic acid (NA) (30 μg), ciprofloxacin (CIP) (5 μg), tetracycline (TE) (30 μg), sulfamethoxazole/trimethoprim (SMX) (25 μg) and gentamicin (CN) (10 μg). The colistin susceptibility assessment was performed in triplicate using Euvsec FRCOL microtitre plates (Thermo Fisher Scientific, Milan, Italy) with concentrations of colistin ranging from 0.12 to 128 mg/L (cut-off > 2 mg/L). *E. coli* ATCC 25,922 and ZTA14/0097EC were used as quality and positive control strains, respectively.

The plates were incubated at 37 °C for 24 h under aerobic conditions after inoculation. The results were evaluated according to the breakpoints established by the European Committee on Antimicrobial Susceptibility Testing (EUCAST) [26], with the exception of sulfamethoxazole, tetracycline, azithromycin and nalidixic acid, for which the breakpoints published by the Clinical and Laboratory Standard Institute (CLSI) were used [27]. ESBL production was performed by the combined disk test with cefotaxime and ceftazidime alone, in combination with clavulanic acid, and confirmed by the microdilution method using Sensititre™ extended spectrum β-lactamase plates (Thermo Fisher Scientific, Milan, Italy), according to the CLSI guidelines [27].

### 2.4. Statistical Analysis

JASP version 14.1 was used to conduct a statistical analysis. Isolates were defined as multidrug resistance (MDR) having resistance to at least three different antimicrobial classes [28]. A descriptive analysis of individual isolates and the overall prevalence of the resistant strain was carried out. A Chi-square test was used to compare the number of susceptible and resistant isolates for each antibiotic. The value of statistical significance was set at *p* ≤ 0.05.

## 3. Results

A total of 100 *E. coli* isolates, one from each bird, were selected on the basis of MALDI-TOF analysis and the phenotypic profile of antimicrobial resistance according to the bird species, as shown in Table 2. Eighty-five and forty-seven *E. coli* isolates showed a significantly high prevalence of resistance (*p* < 0.001) to ampicillin and amoxicillin/clavulanic acid, respectively. Resistance to ciprofloxacin, cefotaxime and ceftazidime presented values of 18%, 17% and 15%, respectively. *E. coli* isolates were found to be highly susceptible to chloramphenicol (94%) and azithromycin (97%). All isolates were susceptible to colistin and imipenem. Seventeen out of the hundred isolates showed a multidrug resistance profile, including the beta-lactam category, and the pattern of resistance to three or four antibiotic classes was the most common (Table 3). Eight out of the hundred isolates were ESBL-producing *E. coli* and seven showed multidrug-resistance profiles (Table 3). *Salmonella* spp. was not isolated.

## 4. Discussion

The scientific community has recently become interested in wild birds due to their role either as possible spreaders of antimicrobial resistance (AR) genes or as sentinels of AR environmental contamination, especially in urban and suburban areas [29]. In this context, studies concerning wild animals as carriers of resistance genes in the ecosystem have been carried out [30].

In our investigation, the most commonly monitored birds were raptors belonging to different orders, all coming from “WildUmbria” rescue centre involved in the recovery of wildlife across Umbria region territory, prevalently composed of rural and forested areas. They usually presented traumatic lesions due to road accidents, hunting, predation and, more rarely, signs of poisoning, similar to reports from other rescue centres [31,32], and no antimicrobial treatment was performed in the centre or before sampling. A significant number of the isolated *E. coli* displayed resistance to ampicillin, followed by resistance to amoxicillin associated with clavulanic acid, regardless of the bird species investigated. These molecules are commonly used in humans [33] and veterinary medicine, both in mammals [34] and poultry, especially for enteric diseases caused by *Clostridium perfringens* [35]. We may assume that our results, which concur with those previously observed by Giacopello et al. [36] and Vidal. et al. [37], could be a consequence of wastewater contamination by zootechnical and urban centres (hospital and municipal sewage), converging in the environment as well as dietary habits of some species of wild birds, such as raptors. Furthermore, it poses a serious risk to public health, especially to the staff in the rescue centres, as a result of not only their close contact with wildlife but also in view of the precautionary measures to be taken.

The presence of *E. coli* strains resistant to cefotaxime and ceftazidime, which are third-generation cephalosporins, is noteworthy. This is probably due to environmental contamination, mainly caused by an anthropogenic source [38,39], even though a potential role played by cat [40] and dog faeces [41] causing pollution in urban areas (streets, meadows, etc.) should not be disregarded. Although the use of cephalosporins is not allowed in poultry, resistance to these molecules has also been reported in industrially and organically raised chickens [42,43] and hens [44]. This could be a consequence of the vertical transmission of resistance genes by breeders due to the typical pyramidal production structure of the poultry sector [45,46]. Moreover, antimicrobial resistance to cephalosporins has also been associated with the unauthorised use of cephalosporins in the hatcheries of some countries [44,47,48]. Thus, possible contamination of environmental soil from farm wastewater cannot be excluded as a possible source of resistance genes for wildlife and vegetation. In contrast to Giacopello et al. [36] and Russo et al. [49], all *E. coli* isolates were proven to be susceptible to imipenem, considered nowadays with third-generation cephalosporins as the first choice for the treatment of serious infections in human medicine [50].

The *E. coli* strains, as expected, were mainly susceptible to azithromycin, a macrolide commonly used in humans [51,52], but not in livestock. The azithromycin-resistant *E. coli* strains were isolated not only from the peregrine falcon, known as a partially migratory bird, but also from the nightjar and the swift, both migratory species and, therefore, able to carry ARG over long distances.

Resistance to other macrolides, such as tylosin and erythromycin, is more frequent since these molecules are largely administered to food-producing animals as they react to different bacterial agents [53,54,55]. It should be emphasized that the use of tylosin in animals can select for resistance to other macrolides [56], posing a serious threat to public health due to its use in human campylobacteriosis [57].

Seven *E. coli* strains, isolated from diurnal raptors (peregrine falcon and buzzards) and passerine birds, were resistant to chloramphenicol, which, although banned in food-producing animals [58], is sometimes used especially in pet animal therapy, where the occurrence of resistance can be observed [59].

All *E. coli* strains were susceptible to colistin, which belongs to the polymyxin group, commonly active against Gram-negative bacteria, including *E. coli* [60]. In contrast, Tolosi et al. [61] recently detected consistent levels of plasmid-mediated colistin resistance genes (*mcr1*–*mcr5*) in environmental samples.

However, though differences exist as regards the “off label” use in the majority of EU community countries, the prevalence of colistin resistance displayed by *E. coli* tends to be low. This could be due to plans based on the “One Health” approach, applied in the EU to control the alarming diffusion of antimicrobial resistance in the animal food chain [62]. On the contrary, a study by Ahmed et al. [63] conducted in Egypt revealed highly prevalent colistin resistance in wild bird *E. coli* isolates, harbouring the *mcr* gene complex, which is known as one of the most responsible for colistin resistance [64]. In this respect, it should be taken into account that Egypt has no regulations to control the use of antimicrobials in animals as growth promoters or to prevent infectious diseases [65].

The phenotypic profile of multidrug resistance to three or four classes of antibiotics was the most common, and beta-lactams were found to be present in all resistance patterns. Eight out of the hundred *E. coli* were ESBL-producing strains, and seven were multidrug-resistant, with the prevalent involvement of raptors, two jackdaws and one swift. These birds are potentially able to contaminate different areas shared with humans, livestock, pets and crops, regardless of their migratory or sedentary habits. The ESBL-producing *E. coli* are known to be responsible for serious infectious diseases in humans and animals [66].

*E. coli* plasmids expressing the ESBL genes are also estimated to produce resistance to other antimicrobial classes, such as aminoglycosides, diaminopyrimidines (trimethoprim) and fluoroquinolones, reducing therapeutic choices in humans [66].

*Salmonella* spp. was not isolated in this study, unlike other studies [67,68,69,70], even though raptors, known to frequently carry this microorganism due to their feeding habits [71,72], were the most investigated species in our study. However, Russo et al. [49] recently reported a low prevalence of *Salmonella* spp. (1.22%) in Southern Italy. Their study supports the hypothesis that the prevalence of *Salmonella* spp. in wild birds depends not only on the bird species but also on the geographical area. Thus, it can be influenced by eating habits and by the level of environmental contamination [49].

## 5. Conclusions

Our results highlight the high presence of commensal *E. coli* strains resistant to ampicillin and amoxicillin associated with clavulanic acid in wild birds, regardless of the species, probably due to environmental contamination. Beta-lactam presence in all multidrug-resistant *E. coli*, including third-generation cephalosporins in some strains, mainly due to sources of environmental contamination of human origin, could be a source of growing concern for public health. The biodiversity of the birds investigated and the difficulties in performing homogeneous sampling do not provide definitive results. However, these outcomes highlight the need to increase monitoring studies in order to better define the role of wild animals in the antibiotic resistance circuit, to determine the potential hazard they have for humans as a source of resistance genes, and to extend samplings to other areas of Italy where anthropization and the presence of livestock settlements could be different.

## Figures and Tables

**Table 1 animals-13-01776-t001:** Order, family, species and number of investigated birds.

Order	Nr	Family	Nr	Species	Nr
Falconiformes	14	Falconidae	14	Eurasian hobby (*Falco subbuteo* Linnaeus, 1758) Eurasian kestrel (*Falco tinnunculus* Linnaeus, 1758) Peregrine falcon (*Falco peregrinus* Tunstall, 1771)	1 10 3
Accipitriformes	19	Accipitridae	19	Eurasian sparrowhawk (*Accipiter nisus* Linnaeus, 1758) Eurasian common buzzard (*Buteo buteo* Linnaeus, 1758)	4 15
Strigiformes	26	Tytonidae	6	Barn owl (*Tyto alba* Scopoli, 1769)	6
		Strigidae	20	Eurasian scops owl (*Otus scops* Linnaeus, 1758) Little Owl (*Athene noctua* Scopoli, 1769) Tawny owl (*Strix aluco* Linnaeus, 1758)	7 8 5
Passeriformes	12	Sturnidae	1	European starling (Sturnus vulgaris Linnaeus, 1758)	1
		Turdidae	7	Blackbird (*Turdus merula* Linnaeus, 1758)	7
		Corvidae	4	Western jackdaw (*Corvus monedula* Linnaeus, 1758) Hooded crow (*Corvus cornix* Linnaeus, 1758)	3 1
Apodiformes	7	Apodidae	7	Common swift (*Apus apus* Linnaeus, 1758)	7
Columbiformes	11	Columbidae	11	Common pigeon (*Columba livia* Gmelin, 1789) Collared dove (*Streptotelia decaocto* Frivaldszky, 1838)	4 7
Bucerotiformes	1	Upupidae	1	Eurasian hoopoe (Upupa epops Linnaeus, 1758)	1
Piciformes	2	Picidae	2	Green woodpecker (*Picus viridis* Linnaeus, 1758)	2
Galliformes	3	Phasianidae	3	Common quail (*Coturnix coturnix* Linnaeus, 1758) Pheasant (*Phasianus colchicus* Linnaeus, 1758)	2 1
Caprimulgiformes	2	Caprimulgidae	2	European nightjar (*Caprimulgus europaeus* Linnaeus, 1758)	2
Pelecaniformes	3	Ardeidae	3	Western cattle egret (*Bubulcus ibis* Linnaeus, 1758)	3
Total	100		100		100

**Table 2 animals-13-01776-t002:** Phenotypic profile of antimicrobial resistance of the 100 *E. coli* isolates according to the species of investigated birds.

Species	Nr	IMP	FOX	FEP	CTX	CAZ	AMP	AMC	CHL	AZM	NA	CIP	TET	SXT	CN	COL	ESBL
Hobby (*Falco subbuteo*)	1	0	0	0	0	0	1	0	0	0	1	1	1	1	0	0	0
Kestrel (*Falco tinnunculus*)	10	0	1	2	2	1	10	4	0	0	0	1	1	1	1	0	1
Peregrine Falcon (*Falco peregrinus*)	3	0	0	0	1	0	3	3	2	1	2	2	2	1	0	0	1
Sparrowhawk (*Accipiter nisus*)	4	0	1	1	2	2	2	2	0	0	0	1	0	1	1	0	1
Buzzard (*Buteo Buteo*)	15	0	0	2	1	1	11	3	1	0	1	2	1	2	1	0	1
Barn owl (*Tyto alba*)	6	0	0	0	0	0	1	1	0	0	0	0	0	0	0	0	0
Scops owl (*Otus scops*)	7	0	0	0	1	1	6	5	0	0	0	2	0	0	4	0	0
Little owl (*Athene noctua*)	8	0	0	0	0	0	7	4	0	0	0	1	0	0	0	0	0
Tawny owl (*Strix aluco*)	5	0	0	1	3	3	4	4	0	0	0	0	0	0	1	0	1
Starling (*Sturnus vulgaris*)	1	0	0	0	0	0	0	0	0	0	0	0	0	0	0	0	0
Blackbird (*Turdus merula*)	7	0	0	0	0	0	7	4	1	0	2	1	1	1	0	0	0
Jackdaw (*Corvus monedula*)	3	0	3	3	3	3	3	3	3	0	3	3	3	3	1	0	2
Hooded crow (*Corvus cornix*)	1	0	0	0	0	0	1	1	0	0	0	0	0	0	0	0	0
Swift (*Apus apus*)	7	0	1	0	2	4	7	6	0	1	1	1	0	0	1	0	1
Pigeon (*Columba livia*)	4	0	0	0	0	0	4	3	0	0	0	1	0	3	0	0	0
Collared dove (*Streptopelia decaopto*)	7	0	0	0	1	0	7	1	0	0	0	1	1	0	0	0	0
Hoopoe (*Upupa epops*)	1	0	0	0	0	0	1	0	0	0	0	1	0	0	0	0	0
Green woodpeker (*Picus viridis*)	2	0	0	0	1	0	2	1	0	0	0	0	0	0	0	0	0
Quail (*Coturnix coturnix*)	2	0	0	0	0	0	2	0	0	0	0	0	2	1	0	0	0
Pheasant (*Phasianus colchicus*)	1	0	0	0	0	0	1	1	0	0	0	0	1	0	0	0	0
Nightjar (*Caprimulgus europeaeus*)	2	0	2	0	0	0	2	1	0	1	0	0	0	0	0	0	0
Cattle egret (*Bubulcus ibis*)	3	0	0	0	0	0	3	0	0	0	0	0	0	0	0	0	0
Total	100	0	8	9	17	15	85	47	7	3	10	18	13	14	10	0	8

IMP: Imipenem; FOX: Cefoxitin; FEP: Cefepime; CTX: Cefotaxime; CAZ: Ceftazidime; AMP: Ampicillin; AMC: Amoxicillin/clavulanic acid; CHL: Chloramphenicol; AZM: Azithromycin; NA: Nalidixic acid; CIP: Ciprofloxacin; SXT: Sulfamethoxazole; CN: Gentamicin; TET: Tetracycline; COL: Colistin.

**Table 3 animals-13-01776-t003:** Multidrug resistance patterns in *Escherichia coli*, including ESBL-producing strains.

	Antimicrobial Classes	Antibiotics	ESBL	TOT *E. coli*
**3**	BETA/QUIN/AMIN	**AMP; AMC**/*CIP*/CN	Neg	2
	BETA/QUIN/TET	**AMP**/*CIP*/TET	Neg	1
	BETA/MACR/QUIN	**AMP; AMC**/AZM/*CIP*	Neg	1
	BETA/TET/SXT	**AMP**/TET/SXT	Neg	1
**4**	BETA/QUIN/TET/SXT	**AMP**/*NA*; *CIP*/TET/SXT **FEP; CTX; AMP; AMC**/*CIP*/TET/SXT **AMP; AMC***/NA*; *CIP*/TET/SXT	Neg Pos (2) Neg	4
	BETA/QUIN/SXT/AMIN	**FEP; CTX; CAZ; AMP; AMC**/*CIP*/SXT/CN **FEP; CTX; CAZ; AMP**/*CIP*/SXT/CN	Pos Pos	2
**5**	BETA/PHEN/QUIN/TET/SXT	**AMP; AMC**/CHL/*NA; CIP*/TET/SXT **AMP; AMC**/CHL/*NA; CIP*/TET/SXT **FOX; FEP; CXT; CAZ; AMP; AMC**/CHL/*NA*; *CIP*/TET/SXT	Neg Pos Pos	3
	BETA/PHEN/MACR/QUIN/TET	**CTX; AMP; AMC**/CHL/AZM/*NA*; *CIP*/TET	Neg	1
**6**	BETA/PHEN/QUIN/TET/SXT/AMIN	**FOX; FEP; CXT; CAZ; AMP; AMC**/CHL/*NA*; *CIP*/TET/SXT/CN	Pos (2)	2

**BETA: Beta-Lactams**; IMP: Imipenem; FOX: Cefoxitin; FEP: Cefepime; CTX: Cefotaxime; CAZ: Ceftazidime; AMP: Ampicillin; AMC: Amoxicillin/clavulanate; **QUIN**: **Quinolones**; NA: Nalidixic acid; CIP: Ciprofloxacin; **AMIN**: **Aminoglycosides**; CN: Gentamicin; **TET: Tetracyclines**; **MACR: Macrolides**; AZM: Azithromycin; **STX: Sulfonamides**; SXT: Sulfamethoxazole; **PHEN: Phenicols**; CHL: Chloramphenicol. In bold—Beta-lactams; italic—Quinolones. Neg: Negative; Pos: Positive. We indicate with the bold and the italic, antimicrobials belong to the same class. Bold are shown all the antimicrobial belonging to Beta-lactams and in italic are reported all the Quinolones.

## Data Availability

Not applicable.

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
