# Peer review of "Antimicrobial Susceptibility of Commensal E. coli Isolated from Wild Birds in Umbria (Central Italy)"

_animals, 2023, doi:10.3390/ani13111776_

Round 1

Reviewer 1 Report

Animals - 2382071

Dear Authors,

I think that the topic of the paper is interesting. Nevertheless, I recommend some modifications before publishing of the paper. In fact, some parts of the manuscript appear as confusing and need to be changed.

Below, I report my remarks for Authors.

Material and methods

A transport medium is mentioned (2.2.2. Line 119) but any information lacks previously. Please, specify sampling and transport conditions (used medium, temperature and transport conditions), in “Sampling” paragraph.

Antimicrobials and concentration disks are mirror-like reported both in table 2 and text (lines 130-137). This sounds as a repetition. Pleas, delete table 2, numbering the following tables accordingly, or the correspondent text.

Results

There is no correspondence between the classifying of animals reported in table 1 (order and family) and table 3 (behaviour: nocturnal and diurnal for raptors, nothing for other species). This leads to confusion in the reading of data. Please, reorganize species in table 3 according to order and family as in table 1.

In table 4, % is mentioned in brackets (first line of the table) but it lacks in the corresponding column where only the number of strains is reported. Please, change table 4 accordingly.

Moreover, reorganize the legend of table 4 more clearly, using BETA: Beta-lactams QUIN: Quinolones, ecc…

Best regards  

Author Response

Referee 1

Material and Methods

A transport medium is mentioned (2.2.2. Line 119) but any information lacks previously. Please, specify sampling and transport conditions (used medium, temperature and transport conditions), in “Sampling” paragraph.

Done

Antimicrobials and concentration disks are mirror-like reported both in table 2 and text (lines 130-137). This sounds as a repetition. Pleas, delete table 2, numbering the following tables accordingly, or the correspondent text.

Thank you for your suggestion, we deleted Table 2 and numbered the following ones accordingly

Results

There is no correspondence between the classifying of animals reported in table 1 (order and family) and table 3 (behaviour: nocturnal and diurnal for raptors, nothing for other species). This leads to confusion in the reading of data. Please, reorganize species in table 3 according to order and family as in table 1.

Table 1 offers a taxonomical overview of the investigated species. The distinction of diurnal and nocturnal raptors in Table 3 was made to support the discussion in which raptors were reported sometimes as nocturnal and diurnal. Thus, to avoid possible confusion in the reading, we modified Table 3 eliminating the behavioral distinction and we grouped following the family and order, although is not specified since it has already reported in Table 1.

In table 4, % is mentioned in brackets (first line of the table) but it lacks in the corresponding column where only the number of strains is reported. Please, change table 4 accordingly.

Done

Moreover, reorganize the legend of table 4 more clearly, using BETA: Beta-lactams QUIN: Quinolones, ecc…

Done.

Reviewer 2 Report

 Materials and methods

 From a practical point of view, how do authors calculate a ratio of 1:10 (line 111) with a cloacal swab?

All the colonies with typical E. coli morphology were selected and confirmed by Maldi-Tof mass spectrometry (line 115): All colonies or a representative isolate from each culture plate? Regarding MALDI-TOF (very briefly) add protocol and data library/es, software/statistical analysis and level of identification.

Transport medium (line 119) has not been described before in the text.

Add reagent manufacturer used for direct agglutination (lines 126-27). This part is confusing because Salmonella spp. was not isolated (line 170).

A reference for Kirby-Bauer disk diffusion method can be added.

Results

Table 3 can be simplified – order antibiotics as in Table 2; there is no need to include S and R. For a bird species and a number of animals just give the R value. Values are reported as percentage – only at the total. The value of the information can be enhanced if main results are highlighted in the table – as a suggestion, indicate MDR isolates using special text (bold letter, or *), etc.   

Intermediate results were not found?

Briefly, describe E coli profile/s after MALDI-ToF analysis.

Also, comment results of ESBL detection using disk test vs microdilution method.

Discussion

The birds sampled in this study come from a wildlife rescue center. Had these birds been treated with antibiotics previously?

Some information about the origin of the animals would be interesting, such as area of origin (nature park / urban?). Also, age can be interesting.

The sentence “the most commonly monitored birds came from the "WildUmbria (line 191)” is not clear – Not all animals came for the recue center? Or the species monitored in the study are frequently used in AMR studies?

Why author assume that wastewater contamination is the origin (line 201) rather than access to urban rubbish, landfills, or other sources? This is suggested later (line 208). Other option is included in the conclusion (lines 273-275).

Paragraph 226-234 can be removed/shortened as this is a discussion of drugs not tested in the report.

Sentences 273-275 and 278-279 are not conclusions of the study.

References

Reference 3: Blanco G et al., Environ. Microbiol. 2007, was retracted. This is not suitable as a reference.

Other issues

Line 53: antimicrobial resistant genes …  antimicrobial resistance genes

Line 57: richly  - I think that the word can be removed

Line 58 trait d'union – better use a simple English word (easy to read)

Line 63: the sea … seawater

Line 81: to be the microorganisms, in which …  - remove the ,

Line 85 and other parts of the text: (ESBL) producing E. coli … (ESBL)-producing E. coli

Lines 130-132: order antibiotics in the text as in table 2.

Line 159, 161, 171, 221: 100 E. coli strains …100 E. coli isolates  

Line 160: was shown … is shown

Line 161 and other parts of the text: Eighty-five (85%) and forty-seven (47%) – there is no need to repeat with numbers.

Line 162-166: No need to repeat antibiotic codes

Line 241: the sentence is long and difficult – try to remove …in a monitoring study…

The three paragraphs at the end of the discussion section contain sentences that are very long.

Reference 4: check format.

Some comments have been included in "other issues".

Author Response

Referee 2

From a practical point of view, how do authors calculate a ratio of 1:10 (line 111) with a cloacal swab?

The cloacal swab was placed in a transport medium and it was vortexed as soon as it arrived in the laboratory. Then, 1 ml of the sample was placed into 9 ml of Buffered Peptone Water (BPW), in order to obtain 1:10 ratio. However, it has been further detailed on line 111.

All the colonies with typical E. coli morphology were selected and confirmed by Maldi-Tof mass spectrometry (line 115): All colonies or a representative isolate from each culture plate? Regarding MALDI-TOF (very briefly) add protocol and data library/es, software/statistical analysis and level of identification.

According your request to give more information to MALDIi-TOFmass spectrometry, we modified the text as follows: ‘All the colonies with typical E. coli morphology were selected and identified using a MALDI-TOF MS instrument (Microflex LT Smart Biotyper with FlexControl Biotyper 3.4 software, Bruker Daltonics, Bremen, Germany). See line 117 -119

Transport medium (line 119) has not been described before in the text.

It has been added, see line 111-112.

Add reagent manufacturer used for direct agglutination (lines 126-27).

It is not necessary anymore, see below.

This part is confusing because Salmonella spp. was not isolated (line 170).

Right. One of our goals was the investigation of Salmonella spp. (see in the objectives of our work) and the following identification, if present. Since Salmonella was not detected, we deleted the identification method, eventually used.

A reference for Kirby-Bauer disk diffusion method can be added.

Done (see line 138)

Results

Table 3 can be simplified – order antibiotics as in Table 2; there is no need to include S and R. For a bird species and a number of animals just give the R value. Values are reported as percentage – only at the total. The value of the information can be enhanced if main results are highlighted in the table – as a suggestion, indicate MDR isolates using special text (bold letter, or *), etc.

Thank you for your helpful suggestion. We deleted S column, whereas the MDR profiles of the isolates are shown in Table 3.

Briefly, describe E coli profile/s after MALDI-ToF analysis.

Thank you for your comment. The isolates were identified as belonging to the E. coli species when they were lactose-fermenters and the score value for identification after MALDI-TOF analysis was above 2. We modified the text accordingly. We add the sentences in Material and Method section (lines 119-121).

Intermediate results were not found?

We did not find intermediate resistant strains.

Also, comment results of ESBL detection using disk test vs microdilution method.

This point has been clarified in the manuscript. According to the Clinical and Laboratory Standards Institute (CLSI), disk susceptibility tests and broth microdilution are valid methods to screen and confirm ESBL production. In particular, disk diffusion has been described as a screening method, then, phenotypic confirmatory testing can be performed by broth microdilution assays using ceftazidime (0.25 to 128 µg/ml), ceftazidime +clavulanic acid (0.25/4 to 128/4 µg/ml), cefotaxime (0.25 to 64 µg/ml), and cefotaxime + clavulanic acid (0.25/4to 64/4 µg/ml).

Discussion

The birds sampled in this study come from a wildlife rescue center. Had these birds been treated with antibiotics previously?

Of course not, as specified in line 100, birds were sampled before any antimicrobial treatment. Moreover, animals were usually found by the staff of the center following citizen reports of injured or distressed animals. Thus, hospital admission has readily needed for accurate diagnosis and treatment, either clinical or surgical.

Some information about the origin of the animals would be interesting, such as area of origin (nature park / urban?). Also, age can be interesting.

Following your precious suggestions, we added some information about Umbria region territory (lines 205-206). WildUmbria Rescue Center is involved in wildlife rescue across Umbria region and most birds came from rural and forested areas whereas a smaller sample was rescued in urban and or peri-urban area.

Age determination, although could be of considerable interest, was out of the work aims, moreover is not always easy to determine age in birds even for experienced ornithologists.

The sentence “the most commonly monitored birds came from the "WildUmbria (line 191 )” is not clear – Not all animals came for the rescue center? Or the species monitored in the study are frequently used in AMR studies?

All birds sampled came from WildUmbria rescue center. Most of the animals were directly brought from the wildlife rescue center staff, just a negligible number, like pigeons (4 animals), were brought from citizens. The frequency of monitored species could not be planned, since it depends on several factors, like geographical area (urban/rural/forest) and bio-ecological features. For instance, species that are more common, like the buzzard are represented in our study more than the less common species like the Eurasian hobby (1 specimen). From this point of view a future goal should be a joint and parallel study with another research group from Northern Italy to highlight possible differences both in wild species populations and in their antimicrobial resistance profiles, as we concluded in our manuscript.

Why author assume that wastewater contamination is the origin (line 201) rather than access to urban rubbish, landfills, or other sources? This is suggested later (line 208). Other option is included in the conclusion (lines 273-275)

Right.  It was too hasty including in urban wastewater all contamination coming from urban center. We tried to give more details.

In line 221, we reported prevalently an urban origin due to rubbish, human wastewater and pet feces, since cefotaxime and ceftazidime, a third generation of cephalosporins, are not used in zootechnical animals 213-216

Paragraph 226-234 can be removed/shortened as this is a discussion of drugs not tested in the report.

Right. We deleted some sentences following your suggestions. However, we would like to leave the concept regarding possible occurrence of cross resistance between antimicrobials belonging to same class

Sentences 273-275 and 278-279 are not conclusions of the study.

Done. Thank you for your right comments.

References

Reference 3: Blanco G et al., Environ. Microbiol. 2007, was retracted. This is not suitable as a reference.

Done.

Other issues

Line 53: antimicrobial resistant genes … antimicrobial resistance genes

Done

Line 57: richly - I think that the word can be removed

Done

Line 63: the sea … seawater

Done

Line 81: to be the microorganisms, in which … - remove the ,

Done

Line 85 and other parts of the text: (ESBL) producing E. coli … (ESBL)-producing E. coli

Done

Lines 130-132: order antibiotics in the text as in table 2.

Based on Reviewer 1 suggestion, Table 2 has been removed.

Line 159, 161, 171, 221: 100 E. coli strains …100 E. coli isolates

Done

Line 160: was shown … is shown

Done

Line 161 and other parts of the text: Eighty-five (85%) and forty-seven (47%) – there is no need to repeat with numbers.

Done

Line 162-166: No need to repeat antibiotic codes

Done

Line 241: the sentence is long and difficult – try to remove …in a monitoring study

Done

Line 162-166: No need to repeat antibiotic codes

Done

The three paragraphs at the end of the discussion section contain sentences that are very long

We tried the best to shorten and simplify the sentences as possible, following your suggestion. We added full stop to shorten sentences to improve the reading of the manuscript.

Reference 4: check format.

Reference 4 has been modified according to the correct format

Reviewer 3 Report

In this MS, the authors investigated the antimicrobial susceptibility of 100 commensal Escherichia coli strains, isolated from wild birds from an Umbrian rescue center and admitted to the Veterinary Teaching Hospital of Perugia (Central Italy) mainly for traumatic injuries. I have some minor specific comments for improvement of the MS as follows:

Lines 1, the title of ‘Antimicrobial Susceptibility of Commensal E. coli isolated from wild birds in Umbria (Central Italy)’ was not quite accurate. It should be corrected as ‘Antimicrobial Susceptibility of Commensal E. coli and Salmonella spp. isolated from wild birds in Umbria (Central Italy)’.

Lines 139-144, please add the breakpoints for each antimicrobial agent.

Lines 153, Are the 100 E. coli strains different or not? Please clearly clarified.

Lines 174-175, ‘Neg’ and ‘Pos’ need to be commented.

Author Response

Referee 3

Lines 1, the title of ‘Antimicrobial Susceptibility of Commensal E. coli isolated from wild birds in Umbria (Central Italy)’ was not quite accurate. It should be corrected as ‘Antimicrobial Susceptibility of Commensal E. coli and Salmonella spp. isolated from wild birds in Umbria (Central Italy)

Actually, we speculated about the potential insertion of Salmonella in the title. However, the presence of antimicrobial susceptibility in the title could be confounding for the reader since there was not any isolation of Salmonella.  We decided to put in the title only commensal E.coli as the main player of our investigation.

Lines 139-144, please add the breakpoints for each antimicrobial agent.

 We inserted the reference by the European Committee on Antimicrobial Susceptibility Testing (EUCAST) and the Clinical and Laboratory Standard Institute (CLSI) for the break point of antimicrobial agent

Lines 153, Are the 100 E. coli strains different or not? Please clearly clarified.

They are different, since E.coli isolates were collected  one for each birds and then selected on the basis of MALDI-TOF analysis, as specified in the Results (line 169)

Lines 174-175, ‘Neg’ and ‘Pos’ need to be commented.

Done as suggested.

Round 2

Reviewer 1 Report

Dear Authors,

the manuscript has been improved. Now, I have only minor remarks:

Material and methods

Table 1, first line: please, change Falco Subbuteo in Falco subbuteo

Results

Table 2: please, correct the scientific name of collared dove: Streptopelia decaopto instead of Streptotelia and use the italic font.